# RecPD: A Recombination-aware measure of phylogenetic diversity

**Cedoljub Bundalovic-Torma**[1], **Darrell Desveaux**[1,2], **David S. Guttman**[1,2]*

**1** Department of Cell & Systems Biology, University of Toronto, Toronto, Ontario, Canada, **2** Centre for the Analysis of Genome Evolution & Function, University of Toronto, Toronto, Ontario, Canada

* david.guttman@utoronto.ca

## Abstract

A critical step in studying biological features (e.g., genetic variants, gene families, metabolic capabilities, or taxa) is assessing their diversity and distribution among a sample of individuals. Accurate assessments of these patterns are essential for linking features to traits or outcomes of interest and understanding their functional impact. Consequently, it is of crucial importance that the measures employed for quantifying feature diversity can perform robustly under any evolutionary scenario. However, the standard measures used for quantifying and comparing the distribution of features, such as prevalence, phylogenetic diversity, and related approaches, either do not take into consideration evolutionary history, or assume strictly vertical patterns of inheritance. Consequently, these approaches cannot accurately assess diversity for features that have undergone recombination or horizontal transfer. To address this issue, we have devised RecPD, a novel recombination-aware phylogenetic-diversity statistic for measuring the distribution and diversity of features under all evolutionary scenarios. RecPD utilizes ancestral-state reconstruction to map the presence / absence of features onto ancestral nodes in a species tree, and then identifies potential recombination events in the evolutionary history of the feature. We also derive several related measures from RecPD that can be used to assess and quantify evolutionary dynamics and correlation of feature evolutionary histories. We used simulation studies to show that RecPD reliably reconstructs feature evolutionary histories under diverse recombination and loss scenarios. We then applied RecPD in two diverse real-world scenarios including a preliminary study type III effector protein families secreted by the plant pathogenic bacterium *Pseudomonas syringae* and growth phenotypes of the *Pseudomonas* genus and demonstrate that prevalence is an inadequate measure that obscures the potential impact of recombination. We believe RecPD will have broad utility for revealing and quantifying complex evolutionary processes for features at any biological level.

## Author summary

Phylogenetic diversity is an important concept utilized in evolutionary ecology which has extensive applications in population genetics to help us understand how evolutionary processes have distributed genetic variation among individuals of a species, and how this

**Data Availability Statement:** All relevant data are within the manuscript and its Supporting Information files. An R library can be downloaded from https://github.com/cedatorma/recpd

**Funding:** This work is funded by a Discovery Grant to DSG from the Natural Sciences and Engineering Research Council of Canada (NSERC) (https:// www.nserc-crsng.gc.ca/index_eng.asp). The funders had no role in study design, data collection and analysis, decision to publish, or preparation of the manuscript.

**Competing interests:** The authors declare that they have no competing interests.

impacts phenotypic diversification over time. However, existing approaches for studying phylogenetic diversity largely assume that the genetic features follow vertical inheritance, which is frequently violated in the case of microbial genomes due to horizontal transfer. To address this shortcoming, we present RecPD, a recombination-aware phylogenetic diversity measure, which incorporates ancestral state reconstruction to quantify the phylogenetic diversity of genetic features mapped onto a species phylogeny. Through simulation experiments we show that RecPD robustly reconstructs the evolutionary histories of features evolving under various scenarios of recombination and loss. When applied to a real-world example of type III secreted effector protein families from the plant pathogenic bacterium *Pseudomonas syringae*, RecPD reveals that horizontal transfer has played an important role in shaping the phylogenetic distributions of a substantial proportion of families across the *P. syringae* species complex. Furthermore, we demonstrate that the traditional measures of feature prevalence are unsuitable as a measure for comparing feature diversity. We also provide a R package implementation of RecPD for public use: https:// github.com/cedatorma/recpd.

## Introduction

The modern genomics era has provided unprecedented opportunities for identifying and quantifying the impact of genetic variants underlying traits of interest, while furthering our understanding of the fundamental evolutionary processes driving the emergence, distribution, and fate of these variants. A critical step in studying these genetic variants is assessing their overall abundance and the distribution of individuals carrying the variants both within and between populations and/or communities. Accurate assessment of these patterns of genetic diversity are essential for linking genotypes to phenotypes and understanding the functional impact of genetic variation. Consequently, it is critical that we have ways to accurately measure and quantify genetic diversity under any evolutionary scenario, including complex distributions brought about through recombination or horizontal gene transfer.

Quantifying diversity is of course not just of interest to those working with genetic data, but has relevance to any discipline where traits, features, or constituents can vary in state. Perhaps the simplest and most common diversity index is abundance (aka frequency or prevalence), which measures the proportion of individuals in a population or community that are of a particular kind, in a particular state, or which carry a trait or feature of interest. Measures of abundance (most commonly used in ecological research) and prevalence (most commonly used in epidemiological research) are often refined or extended to assess *richness* (i.e., the total number of states in a population or community) and *diversity* (i.e., the number of states weighted by their prevalence/abundance). Further, measurements of diversity can be made either within or between populations or communities, with the former being called alpha diversity (e.g., Simpson Index, Shannon Entropy, Hill Numbers) and the latter being called beta diversity (e.g. Jaccard and Sørensen Indices, and Bray-Curtis Dissimilarity) [1–3]. As is clear from the above discussion, diversity can be measured for any type of data that varies across an environment, population or community, including species, operational taxonomic units (OTUs), nucleotides or amino acids, gene families, metabolic capacities, phenotypic traits, or even gene expression levels varying across tissues or cellular environments. In this study, we will simply use the term *feature* to encompass this wide range of data types and define it as any measurable difference between samples.

While prevalence and abundance are intuitive ways to quantify the overall presence / absence of a feature of interest, these measures assume that all samples are independent and uncorrelated. Consequently, they have no ability to assess populations or communities with some underlying structure, such as would be expected when the samples have a shared evolutionary history. Consequently, prevalence and related non-phylogenetic measures can be confounded by complex or unbalanced phylogenetic patterns, complex evolutionary histories, and biased sampling. Phylogenetic diversity (PD) statistics have been developed that incorporate measures of evolutionary relatedness among individuals into the prevalence-based diversity measure discussed above. In general, PD measures are calculated by summing the branch lengths from a common ancestor of a selected group of descendants in a phylogenetic tree representing the total genetic diversity of the sampled population [4–6]. Two of the most widely used phylogenetic diversity metrics are Faith's Phylogenetic Diversity [4] and UniFrac [7]. Faith's PD measures within-population alpha diversity by calculating the sum of the phylogenetic tree branch lengths of all those branches that span the descendants sampled from a given population or sharing the feature of interest. UniFrac measures between-population beta diversity by calculating the proportion of branch lengths in a phylogenetic tree that lead exclusively to the descendants sampled from a given population or that uniquely carry a feature of interest. Both Faith's PD and UniFrac can work with cladograms that only represent the evolutionary branching patterns, or phylograms, where the branch lengths are proportional to evolution time and divergence. When applied to quantifying feature diversity, Faith's PD and UniFrac also implicitly allow for the loss of the feature along phylogenetic lineages (e.g., by pseudogenization); thereby, making them excellent measures for vertically inherited traits. To date, phylogenetic diversity measures have largely been applied to the study of taxonomic diversity, and have been useful for identifying habitats which possess the greatest maximum biodiversity of a particular taxa/species to prioritize conservation efforts [8,9], and studying differences in species communities between habitats or under changing environmental conditions over time [1,2,10].

While phylogenetic diversity measures have proven to be extremely useful, they all share a crucial underlying assumption that the feature of interest is vertically inherited. They can account for the loss of a feature but have no way to correct for non-vertical evolutionary processes such as recombination or horizontal gene transfer. While the assumption of vertical transmission is robust for many systems and studies, it is not appropriate when studying most microbial systems, where the horizontal transmission of genetic material is an important source of genetic novelty, functional innovation, and rapid adaptation.

Here we describe RecPD, a recombination-aware phylogenetic diversity measure. RecPD is uses the same framework as other well-established phylogenetic diversity measures but incorporates methods for dealing with recombination and horizontal transfer. RecPD maps the distribution of a feature of interest onto a species tree and then employs a variety of ancestral state reconstruction approaches to infer the evolutionary histories of gain and loss which may have given rise to an observed distribution of that feature. Through simulation studies we show that RecPD can accurately account for diverse evolutionary scenarios involving recombination, which are ignored by currently available phylogenetic diversity measures. We derive a number of RecPD-based measures that summarize the impact of horizontal transfer, and then show the utility of the approach when analyzing the distribution of type III secreted effector protein families carried by the plant pathogenic bacterium *Pseudomonas syringae*. We also show the broad applicability of RecPD in investigating growth-related phenotypes across the *Pseudomonas* genus.

## Results

### Development of RecPD

The development of RecPD was inspired by the need to understand the distribution of bacterial gene families, so we will discuss the methods from this context although the method is transferable to any other features of interest (e.g., genetic variants, metabolic capabilities, or taxa). We begin with a phylogeny of strains on which we will map the acquisition, loss, descent, and divergence of a gene family of interest. In most circumstances, this phylogeny will be based on the core genome (i.e., those genes found in all strains of the species) and be constructed from the concatenated sequences of core genes. For simplicity, we will refer to this as the species tree. We also have presence / absence distribution of a gene family of interest that varies among the strains in the study set. The goal of RecPD is to determine the phylogenetic diversity of the gene family by reconstructing its evolutionary history on the species tree, accounting for potential horizontal acquisition events that may have occurred.

### Step 1: Assignment of gene family ancestral states on the species phylogenetic tree

To reconstruct the putative lineages where a gene family has arisen during the evolutionary history of a bacterial species, we first begin with ancestral state reconstruction of the gene family. In this case ancestral states considered will be a binary category of gene family presence/absence. To achieve this task, we devised a novel nearest-neighbour (NN) ancestral reconstruction approach (Fig 1A) that begins by identifying which strains in the study set carry the gene family of interest. Based on this, the tips of the species tree are assigned a state of presence or absence for the gene family. Next, each internal/ancestral node is examined and assigned to one of three possible states based on information from the closest-related tips descended from it: 1) 'present' if the nearest-neighbour, i.e., closest related, descendant tips of the node both possess the gene family; 2) 'absent' if the gene family is absent in the nearest-neighbour descendant tips; 3) 'split' if only one nearest-neighbour descendant tip possesses the gene family, which may indicate potential gain/recombination or loss events.

Our framework also allows for the incorporation of other popular approaches for ancestral state reconstruction, such as most-parsimonious reconstruction (MPR) [11] and maximum-likelihood ancestral character estimation (ACE) [12]. The goal of MPR is to find the overall set of internal node states which results in the fewest number of state changes, e.g., most-parsimonious, between ancestral and descendant nodes. The ACE approach was devised as an improvement on MPR, which incorporates branch-length information and inferred gain and loss rates from which the likelihood of given state for each internal node is determined (i.e., from the likelihoods of the states of its descendants).

As the reconstruction of ancestral states is not a trivial task [13,14]. It is important to emphasize that reconstructed ancestral states can vary depending on the approach used, so it is useful to state the potential strengths and limitations inherent to each. The NN approach may produce spurious evolutionary histories of ancestral gene family loss followed by reacquisition depending on the frequency of gain and loss in different lineages. MPR is less liable under these scenarios but does not incorporate divergence between ancestral and descendant nodes; therefore, it may miss certain internal nodes where a state might be present or could find equally parsimonious scenarios resulting in ambiguous state assignments. In the case of ACE, modelling gains and losses as separate processes may resolve ambiguities in some scenarios, but reliably estimating these rates largely depends on the sampling of the given species phylogeny. State assignments will tend to become more uncertain as the amount of divergence between ancestral

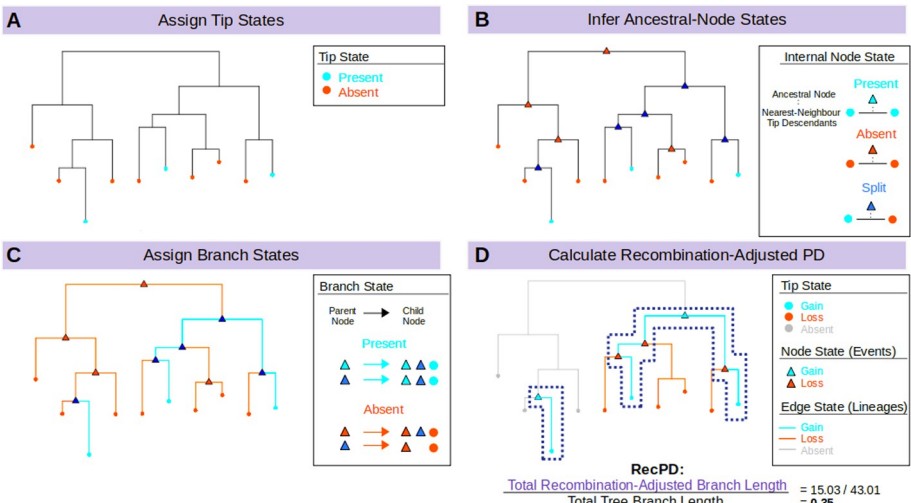

**Fig 1. Outline of the RecPD methodology: Calculating recombination adjusted phylogenetic diversities of feature distributions by employing ancestral state reconstructions.** (A) Tip states are assigned to a given species phylogenetic tree based on presence (teal), or absence (red) of the gene family of interest. (B) Ancestral node states are inferred using one of there ancestral state reconstruction methods and assigned to either presence (teal), absence (red), or split (blue) states, the latter which indicate potential gain/loss events. (C) Branches are assigned to presence states if they join consecutive presence or split nodes/tips (teal), otherwise they are assigned to absence (red). (D) Gene-family lineages are identified, and split state nodes are assigned to gains (teal) or losses (red). Branches descended from the phylogenetic tree root node where no ancestral presence nodes were identified are assigned to absence (grey). RecPD is then calculated as the sum of gene family lineage branch-lengths normalized by the total branch-lengths of the phylogenetic tree.

nodes increases. A combination of approaches could be used in theory to find consensus assignment, although this method has not yet been developed.

## Step 2: Identification of gene family lineages

Ancestral node state assignments made in the previous step are then consolidated to delineate the species lineages and ancestral phylogenetic tree branches where the gene family is likely to have arisen (Fig 1B). This is done by examining all the branches of the species tree and assigning a presence state to branches if their ancestral-descendant node states are also predicted to be present, otherwise a given branch is assigned as an absence state. Some branches may also include nodes where a split/ambiguous state has been assigned as the result of potential gain or loss events occurring in descendant node lineages. In this case, branches are selected for inclusion if the ancestral or descendant node is assigned to the presence state, or both are split states/ambiguous.

After branch-state identification, the final node states are consolidated, with gain and loss event nodes highlighting the particular internal nodes where the gene family appears to have been gained or lost in subsequent descendant lineages respectively (Fig 1C). In addition, the states of tip nodes descended from each gain or loss lineage are updated accordingly.

## Step 3: Calculation of RecPD and nRecPD

After the corresponding gain and loss lineages for a given gene family distribution has been determined, its recombination-adjusted phylogenetic diversity, RecPD, is calculated by summing the total branch-lengths of the gain lineages divided by the sum of the total branch-lengths of the species tree (Fig 1D). RecPD values are bounded to have values ranging from

(~0 to 1), with the lowest values determined by the tip with the smallest branch length from its most immediate parental node (i.e., when feature prevalence = 1), and reaching a maximum when the feature is present at all tips. In addition, the observed values of RecPD for different gene families of the same prevalence could be used to infer the relative impact of recombination on those families, with high RecPD values corresponding to families less impacted by horizontal transfer and low RecPD values corresponding to families heavily impacted by gene gain and loss. Furthermore, normalizing the RecPD of a gene family by its corresponding Faith's PD (nRecPD) yields a more accurate measure which indicates the degree of recombination that has given rise to its distribution: where nRecPD ~ 1 indicates vertical inheritance and nRecPD ~ 0 indicates recently occurring horizontal transfer events.

## Step 4: RecPD-Derived measures

In addition to RecPD, we can also calculate additional measures based on the ancestral mapping of the gene family onto the species tree (summarized with formal definition in Table 1). These measures can be divided into two classes that either assess the overall topological structuring of gene families within the species tree or summarize the evolutionary events influencing gene family lineages (i.e., those lineages of the species tree where the gene family is predicted to be present) inferred by RecPD. The first class of topology-based measures include: *Span*, which measures how much of the maximum possible species diversity is realized for the gene family of interest and is akin to the Faith's PD (S1 Fig); and *Clustering*, which measures the extent to which the gene family lineages identified by RecPD partition among subclades of the species tree, which may be more or less closely related (S2 Fig). While Span and Clustering are conceptually very similar, the former uses branch lengths, while the latter is a cladistic measure that only considers clade structure. The second class of evolutionary event-based measures include: *Longevity*, which is the median normalized evolutionary distance since a gene family was initially gained across species lineages (S3 Fig); and *Lability*, the normalized number of gain, loss, and re-acquisition event nodes occurring across species lineages for a given gene family (S3 Fig).

We also developed a comparative measure for directly quantifying the extent of shared ancestry (i.e., correlation) between different gene families, called RecPDcor. For a pair of gene family distributions, RecPDcor is calculated as the sum of the ancestral lineage branch-lengths

**Table 1. RecPD and derived measures.**

| Name | Type | Description | Formula |
|------|------|-------------|---------|
| **RecPD** | Phylogenetic Diversity | The sum of RecPD-inferred feature gain lineage branch-lengths ($Br_g$) divided by the sum of the total branch-lengths of the species tree ($Br_{tree}$). | $RecPD = \dfrac{\sum_g^G Br_g}{\sum Br_{tree}}$ |
| **nRecPD** | Phylogenetic Diversity | RecPD divided by the sum of branch-lengths descended from feature most-recent common ancestral node ($Br_{MRCA}$), i.e., Faith's PD. | $nRecPD = \dfrac{\sum_g^G Br_g}{\sum Br_{MRCA}}$ |
| **Span** | Cladistic | Faith's PD divided by maximum Faith's PD possible for a feature with equal prevalence ($P$). | $Span = \dfrac{\sum Br_{MRCA}}{max\left(\sum Br_{MRCA}; P\right)}$ |
| **Clustering** | Cladistic | The number tips descended from a gain lineage node possessing a feature ($N_{g:present}$) divided by the total number of descendant tips, divided by the total number of gain lineages identified ($N_G$). | $Clustering = \left(\sum_g^G \dfrac{N_{g(T:present)}}{N_{g(T:present)} + N_{g(T:absent)}}\right) / N_G$ |
| **Longevity** | Event-Based | Median normalized evolutionary distance of all tips possessing a feature to their ancestral gain node ($Br_{g-t_g}$) across species lineages, divided by the maximum root-to-tip distance of the phylogenetic tree ($Br_{root-t}$). | $Longevity = \dfrac{median\left(Br_{g-t_g}; g \in G\right)}{max(Br_{root-t}; t \in T)}$ |
| **Lability** | Event-Based | The number of loss event nodes identified in each feature gain lineage ($N_{g(n:loss)}$), divided by the total number of descendant nodes of that lineage ($N_{g(n)}$). | $Lability = \dfrac{\sum_g^G N_{g(n:loss)}}{\sum_g^G N_{g(n)}}$ |
| **RecPDcor** | Correlation | Jaccard similarity between a pair of RecPD-inferred feature lineages, weighted by their respective branch-lengths. | $RecPDcor_{ab} = \dfrac{\sum Br_a \cap Br_b}{\sum Br_a \cup Br_b}$ |

where they co-occur divided by the sum of branch-lengths from the union of their ancestral lineage reconstructions (S4 Fig). This is in essence the branch-length weighted Jaccard similarity between ancestral gene family lineages.

RecPD accounts for potential recombination events from randomized gene family phylogenetic distributions. As a preliminary exploration of our RecPD method, we present an idealized test-case scenario using a randomly generated species tree of ten tips and assessed the potential impact of recombination inferred from examining all possible gene family presence/absence patterns. We calculated Faith's PD for each gene family phylogenetic pattern to serve as a baseline evolutionary scenario considering only vertical descent, gene family loss, and no recombination. Fig 2A shows that RecPD was affected by the choice of the ancestral state reconstruction method used (NN, MPR, or ACE). When compared to Faith's PD, ACE tends to overestimate PD, which is likely due to increasing uncertainty in state assignments for the deepest ancestral nodes the species tree. In contrast, both the NN and MPR methods consistently predict lower PD values than Faith's, particularly for gene family distributions found in at least half of the species in the tree, with MPR predicting the greatest degree of recombination (Fig 2B and 2C). Therefore, RecPD can serve to identify potential gene family distribution patterns which do not conform to a strictly vertical pattern of inheritance. These results also hold for randomly generated species trees of greater size (S5 Fig).

## Nearest-neighbours ancestral state reconstruction accurately captures recombination events of simulated gene family evolutionary histories

In the real world, we almost never know the true evolutionary histories of a gene family. Although ancestral reconstruction methods are valuable as a means for reconstructing evolutionary histories, more thorough analyses of genomic data are required for support. Therefore,

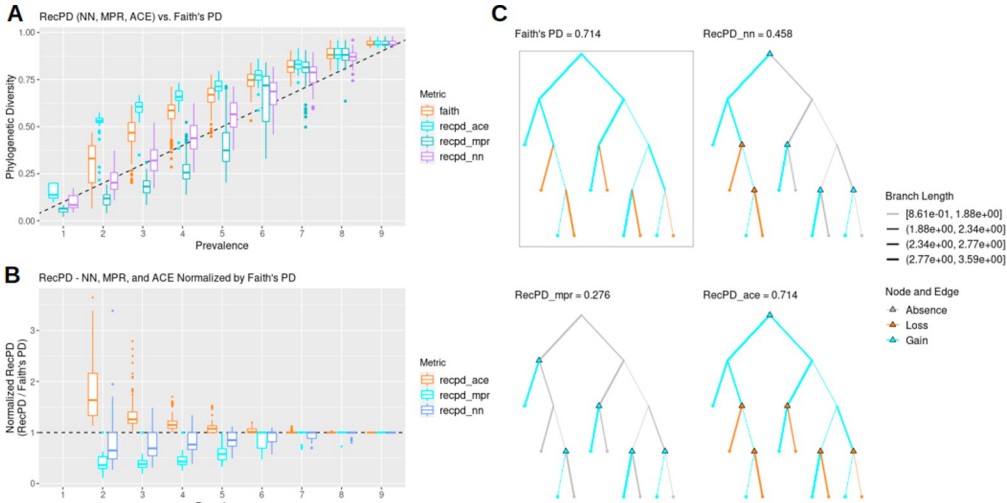

**Fig 2. RecPD results in lower PD estimates compared to recombination-agnostic Faith's PD and is affected by ancestral state reconstruction approach employed.** (A) RecPD using three ancestral state reconstruction methods (NN, MPR, and ACE) and Faith's PD distributions binned by gene family prevalence. Results shown correspond to all 1022 possible randomized gene-family distributions mapped onto a randomly generated example tree topology of ten tips. (B) RecPD normalized by Faith's PD for three ancestral state reconstruction methods binned by gene-family prevalence. Results shown correspond to all 1022 possible randomized gene-family distributions mapped onto a tree of ten tips. (C) Example gene family distribution of prevalence = 5 illustrating differences in inferred evolutionary events using different methods: Faith's PD = 5 losses, 0 gains (boxed), RecPD(NN) = 2 losses, 4 gains, RecPD(MPR) = 0 losses, 5 gains, and RecPD(ACE) = 5 losses, 0 gains.

we simulated gene family histories on randomly generated species trees using a Poisson process to model recombination and loss (Table 2 and S6 Fig). With these 'known' gene family histories we evaluated how accurately the different ancestral state reconstruction approaches, NN, MPR, and ACE, perform in identifying recombination events under diverse evolutionary scenarios, e.g., loss-recombination balanced, loss dominant, or recombination dominant.

To determine the accuracy of RecPD and each ancestral reconstruction approach (NN, MPR, and ACE) we calculated the ratio between the calculated RecPD values, and the known PD based on the simulated distribution. As before, Faith's PD was also calculated as a baseline comparison assuming no recombination. The results of our simulation using a tree of ten tips (Fig 3) show that Faith's and RecPD using ACE result in similar error rates, largely overestimating the PD of gene families, recapitulating results shown previously (mean error Faith = 0.46 ± 0.68; mean error ACE = 0.57 ± 0.78). Conversely, RecPD using MPR underestimated gene family PD (error MPR = -0.18 ± 0.24). Surprisingly, our newly devised NN method was shown to be the most accurate (mean error = 0.083 ± 0.28) in correctly reconstructing gene family histories evolved under high-recombination scenarios. These results were consistent across different evolutionary scenarios (S7 Fig) and for other simulations using randomly generated species trees of ranging from 50 to 100 tips, with the observed error decreasing with increasing tree size (S8 Fig).

RecPD identifies gene family distributions with shared and unrelated evolutionary histories. We next explored the use of RecPD in the context of identifying gene families with shared evolutionary histories. From the RecPD NN ancestral reconstructions for all randomized gene family distributions generated in our first analysis, we calculated the RecPDcor values of their recombination adjusted evolutionary histories, resulting in 521,731 unique pairwise comparisons (note that identical distributions were excluded). In the case of gene family distributions with identical prevalence, we observed that RecPDcor values can vary considerably (Fig 4), particularly for distributions of lower prevalence, reflective of their greater possibility of having evolved through recombination (see Fig 2). Similarly, RecPDcor values tended to be lower for gene family distributions with greater relative differences in prevalence, which is to be expected as the result of limited overlap in evolutionary histories (S9 Fig).

We also compared RecPDcor against other correlation measures to examine the effect of recombination in determining gene family co-occurrence (Fig 4B). The two measures employed were a Faith's PD-based (faith_cor) branch-length weighted Jaccard similarity, which ignores recombination and assumes vertical ancestry of gene families, and Jaccard similarity of co-occurrence among species/tips of the tree (tip_jacc), which ignores gene family ancestries. It was observed that the correlations of gene family distributions tended to be overestimated when not accounting for potential recombination, while ignoring their evolutionary history altogether resulted in their underestimation. In the latter case, by comparing the differences between RecPDcor and tip Jaccard similarities, we could identify several instances where tip Jaccard either missed or identified spuriously correlated gene families (Fig 5).

**Table 2. Gene family evolutionary history simulations: Parameters used.**

| Tree Size (Ntips) | # of Trees (N_trees) | Longevity Rate (long_r) | Loss Rate (death_r) | Recombina-tion Rate (recomb_r) | # Rate Parameter Sets | Simulations per Set (N_iter) | # of Simulations |
|---|---|---|---|---|---|---|---|
| 10 | 10 | 0,1 | 0,2,4,6,8,10 | 0,2,4,6,8,10 | 72 | 25 | 5894 |
| 50 | 10 | 0,1 | 0,2,4,6,8,10 | 0,2,4,6,8,10 | 72 | 25 | 8173 |
| 100 | 10 | 0,1 | 0,2,4,6,8,10 | 0,2,4,6,8,10 | 72 | 25 | 8147 |

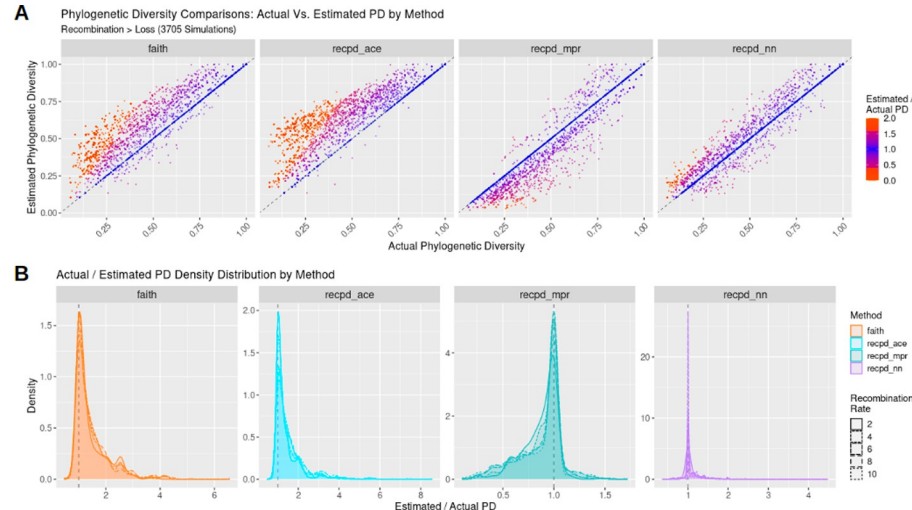

**Fig 3. RecPD with nearest-neighbours (NN) ancestral state reconstruction accurately identifies simulated gene family evolutionary histories, while MPR and ACE over- and under- estimate recombination, respectively.** Summary of simulations of gene family evolution comparing actual phylogenetic diversity to estimated diversity using Faith's PD and RecPD employing three different ancestral reconstruction methods (NN, MPR, and ACE: see Table 1 for parameters used and number of simulations run). (A) Scatterplots of estimated PD values against actual PD values by method. (B) Corresponding density plot distributions. Results are shown for recombination predominant rate regime on randomly generated trees with 10 tips.

## RecPD characterization of *Pseudomonas syringae* type III secreted effector families

*Pseudomonas syringae* is a highly diverse phytopathogenic bacterial species complex that includes over 60 pathogenic varieties that cause many agronomically important crop diseases

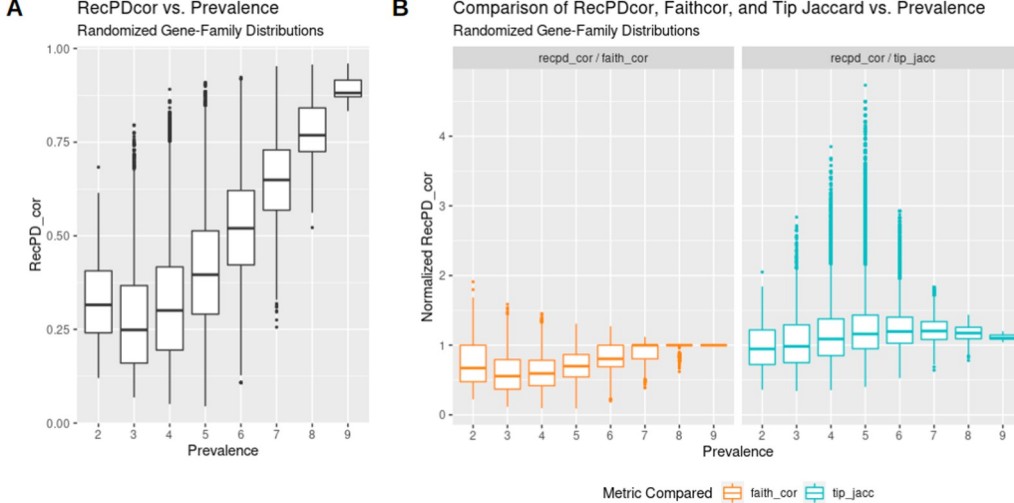

**Fig 4. Pairwise gene family evolutionary history correlations using RecPDcor differ in comparison to recombination-agnostic and phylogeny-agnostic approaches.** (A) RecPD correlation (RecPDcor) values for randomized gene family distributions vs. prevalence reveals that the majority of low-prevalence trait distributions have distinct evolutionary histories; (B) RecPDcor of trait distributions substantially differs when compared to 1) Faith's PD based branch-length weighted Jaccard similarity (recombination-agnostic), and 2) tip presence and absence Jaccard similarity (phylogeny-agnostic) measures: 1) Ignoring recombination results in over-estimation of low-prevalence feature evolutionary history correlations (RecPDcor / Faith's PD < 1); 2) Ignoring evolutionary history results in under-estimation of intermediate-prevalence feature evolutionary history correlations (tip_jacc / RecPDcor > 1).

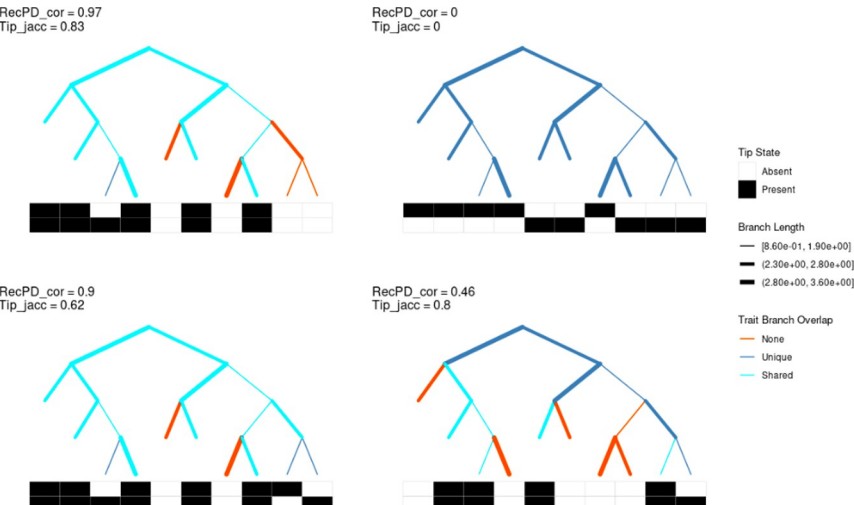

**Fig 5. RecPDcor identifies correlated gene-family distributions missed by tip Jaccard similarity.** Top panels–Distributions of two features (black = present, white = absent) arrayed against a species tree. RecPDcor and tip Jaccard similarities both identify correlated and anti-correlated gene families. Bottom panels–Distributions where RecPDcor reveals correlated gene family distributions where tip Jaccard does not, and where tip Jaccard overestimates correlation of gene families with distinct evolutionary histories. Tree topologies are represented as cladograms with branches of equal length; actual branch-lengths are indicated by branch-thickness as indicated in the legend. Branches are assigned to different categories based on the overlap of their RecPD-inferred gene family lineages: *Shared*—branches where both traits are present (teal); 2) *Unique*—branches where only a single trait is present (blue); *None*–branches where both traits are absent (red).

[15–18]. Decades of research have established *P. syringae* as an important model for the study of host-pathogen interactions. One factor that makes *P. syringae* a particularly adept phytopathogen is its use of a type III secretion system and diverse repertoires of type III secreted effector proteins (hereafter effectors), which have evolved to promote disease by disrupting host immunity and cellular homeostasis [19,20]. In turn, plant hosts have evolved a layer of immunity that triggers when receptors recognize the presence or activity of pathogen effectors [19,21–23]. As a result, the outcome of any particular host-pathogen interaction, and pathogen host specificity in general, largely depends on the specific effectors carried by the pathogen and the specific immune receptors carried by the host. These interactions have led to a co-evolutionary arms race and the accumulation of extensive effector and immune diversity. The *P. syringae* species complex has at least 70 characterized effector families, most of which include numerous diverse alleles that have evolved through both vertical and horizontal evolutionary processes [15]. There is huge diversity in the suites of effectors carried by individual *P. syringae* strains, with most strains carrying ~30 effectors (±9 stderr) [15].

We applied RecPD to a previously published dataset of 529 representative effector alleles distributed among the 70 effector families identified from a collection of 494 sequenced *P. syringe* strains [15,24]. *P. syringae* strains are classified into phylogroups based on their placement in a core genome phylogenetic analysis, with phylogroups varying in overall size and diversity. We mapped effector alleles onto the core genome (i.e., species) tree and found wide variation in the prevalence and distribution of families (Fig 6A) [15]. Relatively few effector families are widely conserved across the *P. syringae* species complex, and effector families of similar levels of prevalence can be distributed in very different ways across the phylogroups, suggesting the importance of recombination and loss in effector evolution [15,24].

We used RecPD to gain a better understanding of effector evolutionary diversity. Not unexpectedly, we find that effector phylogenetic diversity is positively correlated with prevalence,

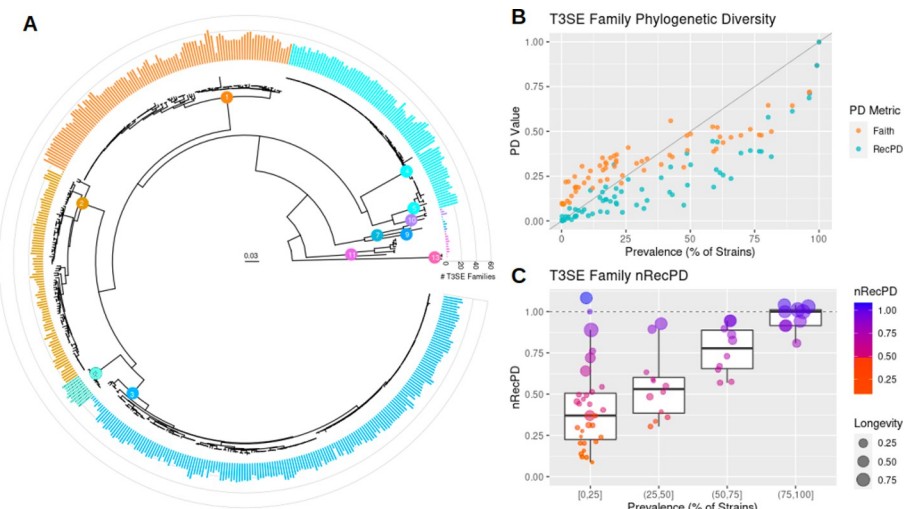

**Fig 6. RecPD reveals significant impact of recombination in the phylogenetic distributions of *P. syringae* effector families.** (A) Core genome phylogeny of the *P. syringae* species complex, with internal tree nodes indicating *P. syringae* phylogroups. The outer ring barplot shows the total number of distinct effector families carried by each strain and coloured according to strain phylogroup designation. (B) Plot of effector family prevalence against RecPD(NN) and Faith PD for all 70 effector families. (C) Effector family RecPD values normalized by Faith's PD, binned by effector family prevalence. The point size indicates effector family longevity.

however we note that relying on prevalence alone gives a misleading view of the phylogenetic distribution of effectors as a whole (Fig 6B). Notably, both Faith's PD and RecPD values were lower on average when compared to effector prevalence, which reflects the impact of shared evolutionary history and the effect of sampling biases inherent in real-world datasets. Furthermore, we also note the importance of considering the effects of recombination when calculating phylogenetic diversity. In the instance of lower-prevalence effector families (found in less than ~ 25% of strains), Faith's PD values tend to be larger than expected based on the observed prevalence, possibly due to impact of horizontal gene transfer that distributes many of these families across multiple phylogroups (i.e., deeper common evolutionary ancestry including branches with greater evolutionary divergence). The impact of horizontal transfer is much more evident from the RecPD values, which are consistently lower than the corresponding Faith's PD values. This is made even more clear when we normalize the PD values by calculating the ratio of RecPD to Faith's PD (nRecPD). Since Faith's PD assumes strictly vertical ancestry, families with a high ratio (~ 1) can be interpreted as evolving by largely vertical evolutionary processes, while a low ratio supports extensive horizontal transmission. We found that the extent of recombination can vary widely for effector families, even when they have very similar prevalence (Fig 6C). The longevity (median normalized evolutionary distance since an effector family was gained) also appears to be correlated to nRecPD. Taken together, this analysis supports the hypothesis that the majority of effector families have experienced extensive horizontal transfer and have been acquired relatively recently during the evolutionary history of *P. syringae*.

To concretely illustrate the impact of horizontal transfer and utility of RecPD, we highlight two examples. The first is the effector families HopS and HopAW (Fig 7A), which have nearly identical prevalence, with HopS found in 114 strains and HopAW found in 116 strains, but dramatically different distributions and inferred evolutionary histories. The Faith's PD values for the two families are 0.152 and 0.198 for HopS and HopAW, respectively. HopS, with a RecPD value of 0.164, appears to largely follow vertical descent from an early acquisition event in the evolutionary history of *P. syringae* and is highly conserved in phylogroup 1 and 6 strains.

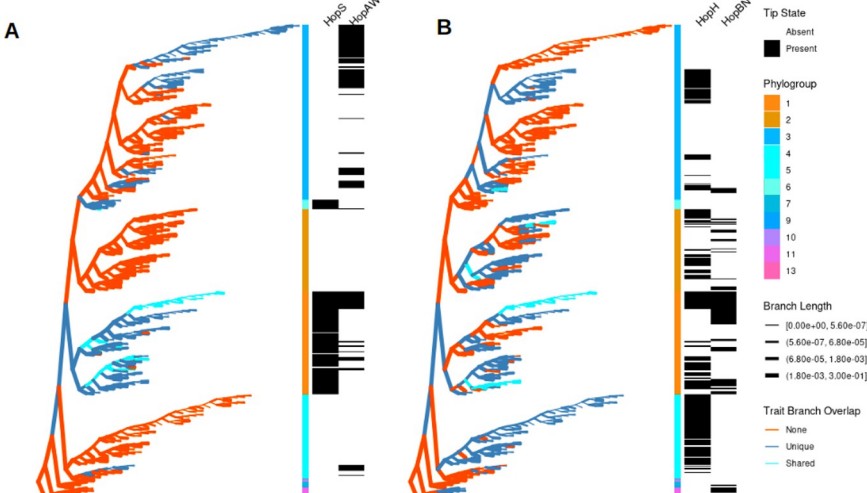

**Fig 7. RecPD gene lineage reconstructions reveal significant differences in evolutionary histories between effector families of similar prevalence and phylogenetic diversity.** Example pairs of effector family distributions mapped onto the *P. syringae* core-genome phylogeny. (A) Effector families HopS and HopAW show similar prevalence (HopS = 114 and HopAW = 116) but different RecPD values (HopS = 0.399 and HopAW = 0.198). (B) Effector families HopH and HopBN show different prevalence HopH = 206 and HopBN = 78) but similar RecPD values (0.16). Tree topologies are represented in a 'willow tree' format, with branches set to equal length, and actual branch-lengths indicated by branch-thickness. Branches are coloured according to overlap between RecPD-inferred gene family lineages.

In contrast, HopAW, with a RecPD value of 0.048, appears to have been gained and lost at numerous times throughout the *P. syringae* phylogroups. The second example is the effector families HopH and HopBN (Fig 7B) which have different prevalence (206 vs. 78 strains for HopH and HopBN respectively), but nearly identical RecPD values of ~ 0.16. Their corresponding Faith's PD values are 0.31 and 0.34, indicating similar overall distribution across the *P. syringae* species phylogeny despite their difference in prevalence. However, by visually comparing their reconstructed evolutionary histories it can be seen that HopH and HopBN have followed dramatically different evolutionary trajectories. The RecPD derived measures also paint a very different evolutionary picture for each family, as shown in Table 3.

## Application of RecPD on *Pseudomonas* spp. growth phenotypes

*Pseudomonas* is a genus within the Gamma-proteobacteria, which includes hundreds of distinct species isolated from a range of diverse environmental habitats [25]. The genus is known for the remarkable metabolic diversity and the broad range of environmental niches which its members inhabit [26], running the gamut from important opportunistic and emergent pathogens of humans and plants, e.g., *P. aeruginosa* [27] and *P. syringae* (as highlighted above) [17,18,28], respectively, to commensal species associated with the plant rhizosphere [29].

**Table 3. RecPD and associated measures for selected effector families shown in Fig 7.**

| T3SE Effector Family | Prevalence | Faith's PD | RecPD(NN) | Span | Clustering | Longevity | Lability |
|---|---|---|---|---|---|---|---|
| HopS | 114 | 0.152 | 0.164 | 0.225 | 0.991 | 0.574 | 0.015 |
| HopAW | 116 | 0.198 | 0.048 | 0.269 | 0.896 | 0.004 | 0.147 |
| HopH | 206 | 0.313 | 0.160 | 0.369 | 0.937 | 0.031 | 0.115 |
| HopBN | 78 | 0.342 | 0.161 | 0.400 | 0.818 | 0.017 | 0.073 |

Species within the genus *Pseudomonas* are now typically delineated via multilocus and genomic methods, although historically they were identified using an array of phenotypic and growth assays [30]. While there are many instances where the phenetic assays disagree with the results from molecular typing, these complementary approaches can provide valuable information about the evolutionary processes driving phenotypic diversification. They also provide an excellent opportunity to apply our RecPD methodology.

We examined the relationships among ten representative *Pseudomonas* species based on the distribution of 18 distinguishing metabolic and phenotypic traits listed in Bergey's Manual of Systematic Bacteriology [31]. The 18 traits have a mean prevalence of 4.3 +/- 2.7 s.d. and fall into approximately four distinct phenotypic clusters that delimit different *Pseudomonas* spp. clades (Fig 8). We used 16S rRNA gene sequences to determine the genetic relationships among the ten bacterial species and applied RecPD to examine the phylogenetic diversity of each phenotype/trait across the *Pseudomonas* species phylogenetic tree. Interestingly, most of phenotype clusters appear to be vertically inherited (nRecPD ~ 1), with only utilization of ketogluconate, L-valine, geraniol, and glucose, and the presence of fluorescent pigments showing signs of horizontal transfer (nRecPD ~ 0.7). These results indicate that most of phenotypes historically used to classify *Pseudomonads* have been transmitted vertically, explaining why they proved to be robust for species delineation.

## Discussion

Diversity measures, such as phylogenetic diversity, have been used to gain valuable insight into the complexity of ecological communities. These statistics can be used to quantify diversity both within and between species or communities (alpha and beta diversity, respectively), and to assess richness (i.e., how many), divergence (i.e., how different), and regularity (i.e., how uniform). Put another way, these measures assess the sum, mean, and variance of the phylogenetic differences among organisms [6]. Despite their tremendous utility, all these measures have a

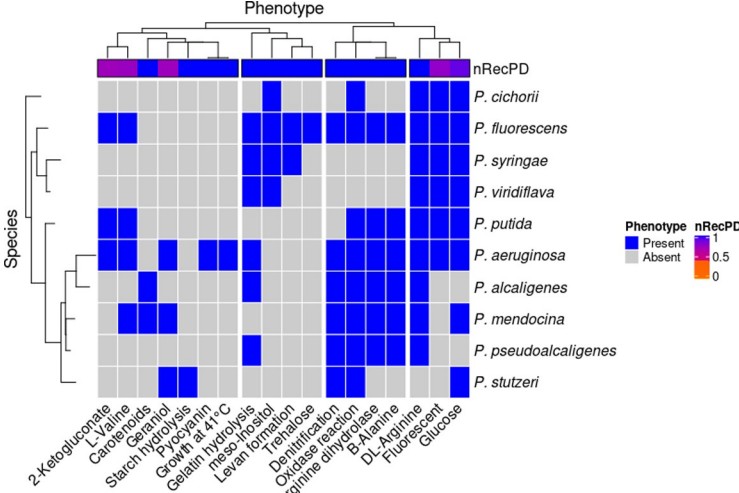

**Fig 8. nRecPD indicates largely vertical descent of *Pseudomonas* spp. growth phenotype distributions, revealing clade specific loss-patterns and between-clade recombination.** A heatmap showing presence / absence profiles of 10 growth phenotypes assayed over 10 representative strains of the genera *Pseudomonas*. Growth phenotype columns are hierarchically clustered and split into 4 major clusters, while species are arranged according to a 16S rRNA phylogenetic tree. Corresponding nRecPD values calculated for each growth phenotype are indicated by the top horizontal annotation strip, with blue indicating vertical descent (nRecPD = 1) and purple indicating signatures of recombination (nRecPD ~ 0.7).

common underlying assumption that raises concerns about their applicability for the majority of life on Earth. Specifically, they assume that organisms under study have descended from a common ancestor strictly through vertical descent. While this may be a reasonable assumption for eukaryotes, it is certainly less valid for bacteria and archaea, where horizontal gene transfer can occur within and between species and dramatically influence their adaptive capabilities.

We developed our recombination-aware phylogenetic diversity metric RecPD to provide a framework for understanding ecological and evolutionary diversity that is robust to the presence of horizontal gene transfer and recombination. RecPD utilizes ancestral state reconstruction to infer the evolutionary histories of features of interest (e.g., gene families, metabolic or phenotypic traits, taxa, etc.) in the context of a given phylogeny. It then identifies evolutionary gains and losses of these features on the tree and quantifies the diversity for only that fraction of the species tree where the feature is present. It then enables the calculation of a number or related measures that quantify the amount of recombination that has occurred in the history of the feature. In general, RecPD provides an intuitive statistic for comparing the diversity and impact of recombination on any feature of interest. It also provides a way to identify lineages that have gained a feature of interest via horizontal transfer, which may otherwise be difficult to determine with features that do not have a clear pattern of descent.

An important step in calculating RecPD is the reconstruction of ancestral states to infer gain and loss events. RecPD can use either of the two well-established ancestral state reconstruction approaches that use distinct modelling frameworks, e.g., parsimony (MPR) and maximum likelihood (ACE). We also introduce a novel approach that is based on nearest-neighbouring states (NN). When evaluating the accuracy of each ancestral reconstruction approach we found that MPR and ACE led to under- and over-estimation of the phylogenetic diversity of simulated gene family histories, respectively, reflective of the assumptions and limitations particular to each modelling framework. We found that the NN ancestral state reconstruction approach provided the most accurate reconstructions, and importantly, performed robustly under evolutionary scenarios of elevated recombination and loss. In addition, we also showed that correlation of ancestral lineages could serve as a useful extension of traditional genomic-context approaches to assess functionally linked gene families [32,33]. However, we caution that the reconstructed gene family histories are still best-guesses given the data at hand, and predicted horizontal transfer events should be used as a starting point for validation using other methods, e.g., conservation of genomic neighbourhood, association with mobile genetic elements, GC content, nucleotide diversity, or species-gene tree topological concordance [34].

The utility of RecPD is clear when analyzing gene families such as *P. syringae* effectors that function as both virulence factors and immune elicitors. These effectors are subject to strong selective pressures, frequent horizontal transfer, and pseudogenization. In a preliminary study of *P. syringae* effectors, we demonstrate that RecPD provides greater insights into effector diversity and evolution than non-phylogenetically aware methods. Interestingly, the dynamic nature of effector evolution through horizontal transfer was sharply contrasted with the largely vertical transmission seen for growth phenotypes of the *Pseudomonas* genus. Taken together, these examples highlight the broad applicability of RecPD for the studying the processes of diversification operating at different scales of biological and evolutionary resolution.

From our simulation experiments and application to real world data, we observed that RecPD performs with little burden on computational resources (for example, the *P. syringae* example above for a tree with 494 species and 70 features took less than one minute, whereas our simulation results for a tree with 100 tips an 8147 features results took approximately 25 minutes). However, in less than ideal situations, important considerations are warranted which will impact the applicability of RecPD. This includes the availability of high-quality

genome assemblies required to construct a reliable species phylogenetic tree, insufficient sampling of species to adequately capture phylogenetic diversity, and incomplete coverage of features for the corresponding species phylogeny. Researchers should take care to assess the reliability of tree topology, particularly by assessing bootstrap support for internal-nodes, which will impact the reliability of lineage reconstructions. Furthermore, RecPD does not take into consideration differential abundance of features, so care must be taken when comparing multiple features of different abundances. An important avenue for future development will be the incorporation of options for abundance-weighted phylogenetic diversity calculation, and ancestral reconstruction approaches for continuous features, which will expand the application of RecPD to metagenomic community surveys.

In addition to quantifying the impact of recombination, RecPD may also be of value in analytical approaches that need to control for population structure, such as genome wide association studies (GWAS). Bacterial GWAS approaches are heavily dependent on population structure corrections that control for the evolutionary history of the sample [35–41]. The ability of RecPD to identify and quantify recombination may increase the power of these statistical methods and provide an interesting avenue for future development. In general, RecPD has great potential for quantifying diversity and assessing the contributions of vertical and horizontal modes of evolution, which is of critical importance for understanding the processes driving the evolution of many bacterial and archaeal gene families

## Methods

### Definition of RecPD

Given a phylogenetic tree and a list representing feature of interest and its presence/absence state for each tree tip, ancestral state reconstruction is performed (see Methods–Development of RecPD–Step 1) and the ancestral states of internal tree nodes and branches are assigned. Ancestral feature lineages are identified by traversing branches from each feature presence state tip to its deepest ancestral presence state nodes in the tree (i.e. presence state nodes immediately descended from an absent state parental node). RecPD is then calculated by the sum of all unique ancestral presence state branches divided by the total branch length of the phylogenetic tree:

$$RecPD = \frac{\sum_{g}^{G} Br_{g}}{\sum Br_{tree}}$$

Where:

- $Br_{g}$- all presence state branch lengths between the deepest ancestral tree presence ($g$) nodes to directly descendant feature presence tips.

- $Br_{tree}$ $All$ branch lengths of the phylogenetic tree.

### RecPD Development and implementation

All code development, final implementation of RecPD analyses, simulation experiments and figures presented in this work was performed in RStudio (R version 4.0.2) [42]. The ape library [43] was used for basic phylogenetic tree import and processing tasks, generation and visualization of random trees used in methods development section, and ancestral reconstruction using MPR and ACE. Faith's PD was calculated using the pd() function from the picante library [44]. The ggtree library [45] was used for final phylogenetic tree visualizations. R code for running RecPD can be found in S1 File.

### Gene family evolutionary simulation

Simulation gene-family history evolution was performed using a Poisson process to model gene family recombination and loss events onto a provided species tree phylogeny (code supplied in S2 File). The method is motivated by ideas from the modelling of birth-death phylogenetic trees [46], similar approaches used for simulating microbial gene-tree phylogenies [47], and the simulation of bacterial genomes and phenotypic evolution on phylogenetic trees [40]. In essence, it models the evolution of a trait (gene-family, variant, or locus) through loss and recombination occurring along the different lineages of a provided species phylogenetic tree. The tree can be either ultra-metric or non-ultrametric, with branch-lengths representing either time since emergence from a common ancestor, or a molecular evolutionary distance (average expected nucleotide/amino acid substitutions per site).

The model requires two parameters, specifying the rate exponents for each type of event and their values can be scaled according to the maximum root-to-tip death of the phylogenetic tree (in our case this value is scaled to 1):

- Extinction/Death/Loss Rate (Er): losses of a locus/state

- Recombination Rate (Rr: gains of a locus/state from one species lineage to another

These rates can be thought of as a summary of the evolutionary selective pressures acting to maintain a locus/state or its selective advantage helping to propagate it. Note that these rates remain constant throughout the evolutionary history of the species tree, however in reality they are likely to vary under different population bottlenecks or changing environments.

Using these rates the Poisson interarrival time distribution for any event (loss or transfer) can be calculated using the exponential distribution with the rate parameter equal to the sum of the extinction and recombination rates: $P(t_i) = (Er + Rr) * \exp(-(Er + Rr)*t_i)$, where exp is the exponential function and $t_i$ is the given inter-arrival time between successive events.

Another important feature of this model is that probabilities of events occurring at a given inter-arrival time:

- Extinction Probability: Er / (Er + Rr)

- Recombination Probability: Rr / (Er + Rr)

In our modelling procedure, first the emergence time of the trait is randomly drawn from a uniform distribution and then randomly assigned to a species lineage existing at that time (trait origination event). Next, the occurrence of events is modelled using a Poisson process by randomly drawing a sequence of interarrival times from the inverse cumulative probability function of the exponential distribution: $e_t = -\log(1-P(i))/(Er + Rr + Lr)$, where i is a random variable sampled from a uniform distribution taking values from [0–1].

The inter-arrival time sequence is cumulatively summed and then added to the emergence time of the first event (cumsum(e_t) + birth_time) which gives a sequence of the the event occurrence times that will then be randomly mapped upon the species tree lineages. Note, only those event occurrence times time of emergence for the locus/state until the time when the species tips are observed (= 1) are considered.

Iterating successively through the event timings, a number between [0–1] is randomly drawn from a uniform distribution (prob_event) and used to determine whether the given event is a loss or recombination:

- Extinction/Loss: if prob_event < = Er / (Er + Rr); otherwise

- Recombination: prob_event $>$ Er / (Er + Rr)

In addition, a locus/state longevity rate (Lr) parameter can be incorporated, which will result in the inter-arrival event distribution of exp(-(L + Er + Rr)), but also add the possibility of no events occurring in the evolution of the trait:

- No Event (Trait State Maintained): if prob_event $<$ = Lr / (Lr + Er + Rr); otherwise

- Extinction/Loss: prob_event $<$ = Er / (Lr + Er + Rr); otherwise

- Recombination: prob_event $<$ = Rr / (Lr + Er + Rr)

If the event is a trait loss, species lineages possessing the locus/state at the given time are extracted and assigned as a loss event, and all descendant lineages occurring after the event time are also assigned as losses. If the event is a recombination, then a species lineage lacking the locus/state at the given time (if it exists) is randomly selected and assigned as a locus gain event, and its descendants are assigned as a locus gain event, in distinction to the initial locus/state origination event. In effect this generates a locus/state distribution presence/absences for the tips of the species tree, as well as the ancestral evolutionary histories of these traits. There may also be tips which never possessed the locus in their evolutionary history (absence). Using this approach, we can examine how frequently locus/state distributions overlap between different evolutionary regimes, e.g. loss dominated vs. recombination dominated.

### *Pseudomonas syringae* type III effector family analysis

Data for *P. syringae* effectors including NCBI genomic accession numbers for 494 *P. syringae* strains used for effector identification, associated strain metadata, and classified effector family sequences originates from a previously published study [15]. Genomic assemblies used were generated using the protocol outlined in [15]. Annotation of genome assemblies was performed using prokka (version 1.14.16) [48], pangenome analysis and core-genome nucleotide alignment was produced using PIRATE (version 1.0.4) [49], from which a core-genome phylogenetic tree was generated using IQ-TREE (version 1.6.12) [50]. All software was run in Linux. Generation of effector presence/absence matrices, RecPD analyses, and phylogenetic tree visualization and annotation were performed in R.

### *Pseudomonas* growth phenotype analysis

Growth phenotypic data for *Pseudomonas* species was extracted from [31] and converted into binary presence / absence format. Corresponding representative species 16S rRNA gene sequences were downloaded from NCBI (including *Cellvibrio japonicus* as an outgroup according to [25] and aligned using MUSCLE (v3.8.1551) [51]. Phylogenetic tree construction was performed using IQ-TREE (version 1.6.12) [50]. Heatmap visualization was generated using the ComplexHeatmap package in R [52].

## Supporting information

**S1 Fig. Illustration of RecPD Span metric calculation with example distributions of identical prevalence but differing Spans.** (A) Span is calculated by summing of branch-lengths joining tips in the phylogenetic tree possessing a gene family with a given level of prevalence. (B) Normalizing by the maximum possible sum of branch-lengths found at the same level of prevalence. (C) Example gene family distributions of prevalence = 4 mapped onto a tree of 10 tips, with maximum, minimum and median span values.
(TIF)

**S2 Fig. Illustration of RecPD Clustering metric calculation with example distributions.** Clustering is calculated from the sum of the number of internal presence state nodes identified, normalized by the maximum clustering possible based on tip prevalence.
(TIF)

**S3 Fig. Illustration of RecPD Longevity and Lability metric calculation with example distributions of identical prevalence but different Longevity and Lability.** Longevity is calculated as the median branch-lengths of ancestral gain to loss internal nodes and presence state tips, normalized by the maximum root-to-tip distance of the phylogenetic tree. Lability is the corresponding sum of ancestral gain and loss nodes identified for each RecPD reconstructed gene-family lineage divided by the total number of gained and lost tips. Panels A–C show gene-family distributions of prevalence = 4 mapped onto a tree of 10 tips having approximately equal Longevity and Lability (A), High Longevity and low Lability (B) and low Longevity and high Lability (C).
(TIF)

**S4 Fig. Illustration of RecPDcor metric calculation.** A pair of RecPD gene family reconstructions are merged into a consolidated phylogenetic tree with mutually present (teal), unique (blue), and mutually absent (red) ancestral branches identified. RecPDcor is then calculated as the sum of branch-lengths mutually present branches divided by the total sum of mutually present and unique branches.
(TIF)

**S5 Fig. Differences of RecPD by NN, MPR, and ACE ancestral reconstruction approaches normalized by Faith's PD: Random gene family distributions for trees of 100, 500, and 1000 tips.** 50 random gene-family distributions were generated at each level of prevalence indicated, resulting to 451 distributions in total for each tree.
(TIF)

**S6 Fig. Gene family evolutionary history simulation—outline of simulation experiment protocol.**
(TIF)

**S7 Fig. RecPD NN, MPR, and ACE vs. Faith's PD–estimated / actual PD for evolved gene family distributions by rate regime (example trees with 10 tips).**
(TIF)

**S8 Fig. RecPD values for simulated gene family histories shows consistent performance for trees of different size.** Difference of NN, MPR, and ACE and Faith's PD compared to actual PD for evolved gene family distributions by rate regime. (A) Trees with 50 tips. (B) Trees with 100 tips.
(TIF)

**S9 Fig. Effect of feature prevalence differences on corresponding RecPDcor values.** Facets represent the distribution of RecPDcor values binned by the normalized prevalence differences, min (prevalence) / max(prevalence), of each pairwise randomized gene-family distribution comparison compared. Note, normalized prevalence difference = 1 indicate distributions with identical prevalence. Results correspond to a test-case of all possible 1022 gene-family distributions mapped onto a tree of 10 tips.
(TIF)

**S1 File. R code for calculation of RecPD and associated metrics.**
(R)

**S2 File. Tutorial for running RecPD analyses.**
(HTML)

**S3 File. R code for gene family evolutionary history simulations.**
(HTML)

## Acknowledgments

We would like to thank all members of the Guttman and Desveaux labs for their insight input to this project.

## Author Contributions

**Conceptualization:** Cedoljub Bundalovic-Torma, Darrell Desveaux, David S. Guttman.

**Data curation:** Cedoljub Bundalovic-Torma.

**Formal analysis:** Cedoljub Bundalovic-Torma.

**Funding acquisition:** David S. Guttman.

**Investigation:** Cedoljub Bundalovic-Torma.

**Methodology:** Cedoljub Bundalovic-Torma, David S. Guttman.

**Project administration:** David S. Guttman.

**Resources:** David S. Guttman.

**Software:** Cedoljub Bundalovic-Torma.

**Supervision:** David S. Guttman.

**Validation:** Cedoljub Bundalovic-Torma.

**Visualization:** Cedoljub Bundalovic-Torma.

**Writing – original draft:** Cedoljub Bundalovic-Torma, David S. Guttman.

**Writing – review & editing:** Cedoljub Bundalovic-Torma, Darrell Desveaux, David S. Guttman.

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
