## [Decision Letter · Decision Letter 0]

24 Nov 2021

Dear Guttman,

Thank you very much for submitting your manuscript "RecPD: A Recombination-Aware Measure of Phylogenetic Diversity" for consideration at PLOS Computational Biology.

As with all papers reviewed by the journal, your manuscript was reviewed by members of the editorial board and by several independent reviewers. In light of the reviews (below this email), we would like to invite the resubmission of a significantly-revised version that takes into account the reviewers' comments.

Both reviewers are positive about the value of the proposed measure and the manuscript.

However, they also raise important questions related to the presentation. It would be clarifying to present more context for practical use (particularly, as reviewer #1 points out, because RecPD is presented a very generic framework with several potential uses, but only one scenario is fully explored in the example). Additionally, including a more formal definition of the measure would enable interested readers to understand exactly what is being computed. Furthermore, a discussion of its mathematical properties (see for example, the question from Reviewer #1 on whether it forms a metric) would be warranted and, again, provide context for potential users.

We also echo the reviewer's suggestion that the authors provide an R package (or similar) to increase the usage of the metric in the community.

We cannot make any decision about publication until we have seen the revised manuscript and your response to the reviewers' comments. Your revised manuscript is also likely to be sent to reviewers for further evaluation.

Sincerely,

Luis Pedro Coelho

Associate Editor

PLOS Computational Biology

Kiran Patil

Deputy Editor

PLOS Computational Biology

Both reviewers are positive about the value of the proposed measure and the manuscript.

However, they also raise important questions related to the presentation. It would be clarifying to present more context for practical use (particularly, as reviewer #1 points out, because RecPD is presented a very generic framework with several potential uses, but only one scenario is fully explored in the example). Additionally, including a more formal definition of the measure would enable interested readers to understand exactly what is being computed. Furthermore, a discussion of its mathematical properties (see for example, the question from Reviewer #1 on whether it forms a metric) would be warranted and, again, provide context for potential users.

We also echo the reviewer's suggestion that the authors provide an R package (or similar) to increase the usage of the metric in the community.

Reviewer's Responses to Questions

**Comments to the Authors:**

Reviewer #1: The authors present RecPD, a family of phylogenetic diversity measures that account for putative recombination and horizontal gene transfer events. In my opinion, this is topical work on a relevant topic, with a simple and elegant underlying idea. This is also one of the best written and most accessible manuscripts I've reviewed for a while – congratulations on this beautiful work!

The theoretical arguments and simulations for benchmarking seem reasonable and are well laid-out (with the caveats discussed below). The analysis of real data on P. syringae is very appropriate as an example use case for potential users.

That said, I have a few comments that I feel should be addressed prior to publication.

- While the writing is overall very clear and easy to follow, the terminology and language around "diversity" in the ecological sense is not always accurate. Local ('alpha') and between-community ('beta') diversities are discussed in the Introduction section, but there are no clear definitions of either, and it may be difficult for non-expert readers to place the RecPD family of measures among existing indices (beyond the comparison to Faith's PD).

- Moreover, the authors introduce RecPD as "ecological" diversity measure, but the discussed use cases (both on simulated and real data) are not designed that way: rather than comparing samples of entire communities (e.g. of closely related strains), the authors compare lineage genomes (or traits) directly, as would be done in a comparative genomics study, basically using PD and RecPD as summary statistics on the trait. The authors address this by introducing RecPD as a measure on “features” that can be many things, but it remains unclear how RecPD would be used as an “ecological” diversity measure in the stricter sense in practice.

- Admittedly, while the P. syringae results give a good general example of RecPD’s usefulness, the manuscript does not leave me with a clear idea of how RecPD would be used in practice (and where it would not be appropriate). I believe that the authors should at least outline possible limitations with regards to data types and requirements: can RecPD handle imperfect phylogenies or missing data? How does the method scale computationally to larger problems?

- RecPD and derived measures do not adjust for differential abundance of entities (taxa) carrying ‘features’. As described in the text, RecPD is an adjusted richness measure, but does not account for the frequency with which each trait/feature is observed in a community. I believe this is a relevant limitation of the method that should at least be addressed in the text – also to put RecPD in context of existing measures, see previous point.

- Related to this, a more formal description of RecPD would be desirable. While the text and Fig 1 lay out the concept very well, it would be good to have more exact mathematical formulations, probably in the Methods section.

- Moreover, while the simulations provide some intuition of how RecPD values behave relative to Faith’s PD, the interpretation (or indeed, interpretability) of absolute RecPD values remains unclear. How do these measures behave under different scenarios? Is RecPD bounded, what are extreme scenarios that would provide extreme values? Does RecPD satisfy the properties of a true metric (I believe it does not), does it satisfy the doubling (or replication) principle (I also believe not)? These points should at least be addressed as potential caveats.

- While the authors provide all code as supplement, I strongly encourage them to formally release RecPD as an R package on a public repository to foster its adoption by the community (and further development).

- I also strongly encourage the authors to revise their figures with a view to the use of colourblind-friendly palettes.

Reviewer #2: In the manuscript “RecPD: A Recombination-Aware Measure of Phylogenetic Diversity” the authors propose a diversity metric that accounts for recombination and generally patterns of gene gain/loss. The authors essentially expand upon the useful metric of Faith’s PD to infer ancestral states at nodes, and then to calculate the time in the tree of a gene gain or loss.

This is a clever paper that proposes a novel metric that has good potential to be of use to the microbial ecology and microbial genetics broadly. I appreciated the modularity of the developed method, that it can take ancestral state reconstructions derived from different methods for comparison. I also appreciated that this metric is very simple and intuitive. That is a major strength.

I strongly recommend the authors make an implementation of RecPCD that is available to the academic community, such as through an R package, for example. There are so many different diversity statistics out there. Making RecPD an accessible statistic to calculate (through a package or python module) could make the difference between this being a widely used metric, or one that is not.

In general, the figure legends were not detailed enough to explain the figures. For this manuscript specifically I would recommend including the major takeaways for the figure in the legend. Otherwise it is difficult to understand to what the reader should pay attention in the simulation results.

Minor comments:

Line 64: Should read “distributions of a substantial…”

Line 80: Should read “individuals that carry”

Line 168: Should read “allow for the incorporation”

Line 219: I don’t see Clustering referenced in Figure S2.

Figure 4: The caption for this figure should be expanded to explain major takeaways from figure.

Line 316: Should read “has at least…”

Line 323: Should read “based on their…”

Line 334: Sentence starting with “Notably” is too long.

Lin 771: I don’t understand the statement “Branches are coloured according to overlap between RecPD-inferred gene family lineages”.

Line 428: Should read “In addition to quantifying…”

Line 492: Should it read “that will then be randomly mapped”?

**Have the authors made all data and (if applicable) computational code underlying the findings in their manuscript fully available?**

Reviewer #1: Yes

Reviewer #2: Yes

PLOS authors have the option to publish the peer review history of their article (what does this mean?). If published, this will include your full peer review and any attached files.

Reviewer #1: **Yes: **Thomas SB Schmidt

Reviewer #2: No
---

## [Decision Letter · Decision Letter 1]

7 Feb 2022

Dear Guttman,

We are pleased to inform you that your manuscript 'RecPD: A Recombination-Aware Measure of Phylogenetic Diversity' has been provisionally accepted for publication in PLOS Computational Biology.

Best regards,

Luis Pedro Coelho

Associate Editor

PLOS Computational Biology

Kiran Patil

Deputy Editor

PLOS Computational Biology

Reviewer's Responses to Questions

**Comments to the Authors:**

Reviewer #1: I have no further comments, congrats to the authors on the revision.

Reviewer #2: The authors have addressed the comments from the previous review. I am satisfied with the current manuscript.

**Have the authors made all data and (if applicable) computational code underlying the findings in their manuscript fully available?**

Reviewer #1: Yes

Reviewer #2: None

PLOS authors have the option to publish the peer review history of their article (what does this mean?). If published, this will include your full peer review and any attached files.

Reviewer #1: **Yes: **Thomas SB Schmidt

Reviewer #2: No

---

## [Editor Report · Acceptance letter]

16 Feb 2022

PCOMPBIOL-D-21-01774R1 

RecPD: A Recombination-Aware Measure of Phylogenetic Diversity

Dear Dr Guttman,

I am pleased to inform you that your manuscript has been formally accepted for publication in PLOS Computational Biology. Your manuscript is now with our production department and you will be notified of the publication date in due course.

With kind regards,

Agnes Pap
